# Rapid Detection of Carbapenemases Using NG-Test^®^ CARBA 5 in Positive Blood Cultures: A Diagnostic Test Study

**DOI:** 10.3390/antibiotics13111105

**Published:** 2024-11-20

**Authors:** Diana Munguia-Ramos, Luis Fernando Xancal-Salvador, Verónica Esteban-Kenel, Narciso Ortiz-Conchi, Ricardo Antonio Jaimes-Aquino, Miguel Mendoza-Rojas, Axel Cervantes-Sánchez, Steven Méndez-Ramos, Hector Orlando Rivera-Villegas, Sandra Rajme-Lopez, Karla Maria Tamez-Torres, Carla Marina Roman-Montes, Areli Martínez-Gamboa, Miriam Bobadilla del-Valle, Jose Sifuentes-Osornio, Alfredo Ponce-de-Leon, Maria Fernanda Gonzalez-Lara, Bernardo Alfonso Martinez-Guerra

**Affiliations:** 1Department of Infectious Diseases, Instituto Nacional de Ciencias Médicas y Nutrición Salvador Zubirán, 15 Vasco de Quiroga, Belisario Domínguez Secc 16, Tlalpan, Mexico City 14080, Mexico; dianamunguia@iner.gob.mx (D.M.-R.); hector.riverav@incmnsz.mx (H.O.R.-V.); sandra.rajmel@incmnsz.mx (S.R.-L.); karla.tamezt@incmnsz.mx (K.M.T.-T.); carla.romanm@incmnsz.mx (C.M.R.-M.); luis.ponceg@incmnsz.mx (A.P.-d.-L.); fernanda.gonzalezl@incmnsz.mx (M.F.G.-L.); 2Clinical Microbiology Laboratory, Department of Infectious Diseases, Instituto Nacional de Ciencias Médicas y Nutrición Salvador Zubirán, 15 Vasco de Quiroga, Belisario Domínguez Secc 16, Tlalpan, Mexico City 14080, Mexico; fernando.xancals@incmnsz.mx (L.F.X.-S.); veronica.estebank@incmnsz.mx (V.E.-K.); narciso.ortizc@incmnsz.mx (N.O.-C.); ricardo.jaimesa@incmnsz.mx (R.A.J.-A.); miguel.mendozar@incmnsz.mx (M.M.-R.); axel.cervantess@incmnsz.mx (A.C.-S.); steven.mendezr@incmnsz.mx (S.M.-R.); rosa.martinezg@incmnsz.mx (A.M.-G.); miriam.bobadillav@incmnsz.mx (M.B.d.-V.); 3General Direction, Instituto Nacional de Ciencias Médicas y Nutrición Salvador Zubirán, 15 Vasco de Quiroga, Belisario Domínguez Secc 16, Tlalpan, Mexico City 14080, Mexico; jose.sifuenteso@incmnsz.mx

**Keywords:** carbapenemase, blood culture, carbapenem resistance, point-of-care testing

## Abstract

Background: Infections due to carbapenem-resistant Gram-negative bacteria are emerging as an important challenge in health-care settings and a growing concern worldwide. Lateral flow immunoassay NG-Test^®^ CARBA 5 can detect the five most reported carbapenemases (KPC, OXA-48-like, VIM, IMP, and NDM). Direct testing of positive blood cultures could reduce time to detection. This study aims to validate and report on the diagnostic yield of a novel method for carbapenemase detection in positive blood culture vials using NG-Test^®^ CARBA 5. Methods: We implemented an investigator-developed method for the direct testing of positive blood cultures using NG-Test^®^ CARBA 5. We compared results between genotypic, phenotypic, and direct NG-Test^®^ CARBA 5 in blood. Results: A total of 32 isolates were tested (21 Enterobacterales and 11 *Pseudomonas aeruginosa*). Genotypic testing detected 23 carbapenemases. When comparing the results of NG-Test^®^ CARBA 5 in blood with genotypic testing, agreement was observed in 31/32 (97%) tests. The sensitivity, specificity, positive predictive value, and negative predictive value of the NG-Test^®^ CARBA 5 in blood were 93%, 100%, 100%, and 94%, respectively. Conclusions: Our method using NG-Test^®^ CARBA 5 directly in blood culture samples presented an excellent diagnostic yield when compared to genotypic profiling and permits an accurate detection of carbapenemases.

## 1. Introduction

Antimicrobial resistance remains a leading cause of death worldwide [1]. Infections due to carbapenem-resistant Gram-negative bacteria (CR-GNB) are emerging as an important challenge in health-care settings and a growing concern worldwide. Among Enterobacterales, up to 3.3% of clinical isolates have been reported to be nonsusceptible to meropenem [2]. In Latin America, carbapenem resistance has been widely reported as a leading cause of morbidity and mortality [3,4,5,6,7]. Infections due to CR-GNB are more frequent in those who have recently received broad-spectrum antibiotics, were hospitalized, or underwent surgical procedures, with invasive mechanical ventilation, nursing home residents, and comorbid or immunocompromised patients [8,9,10]. Although variable, mortality can reach 50% in patients with bloodstream infections due to CR-GNB [11,12].

Different carbapenem resistance (CR) mechanisms have been described. Identification of the causative CR mechanism has a direct impact in appropriate antimicrobial selection [13,14]. In Enterobacterales, carbapenemase production is the most frequent CR mechanism, whereas decreased membrane permeability predominates in *Pseudomonas aeruginosa*. Carbapenemases are a group of beta-lactam-hydrolysing enzymes conferring resistance to one or various carbapenems. Both serine carbapenemases (e.g., KPC, GES, OXA-51, OXA-48) and metallo-beta-lactamases (MBLs) (e.g., NDM, VIM, IMP) have been detected in distinct isolates. Although KPC is the most widely distributed carbapenemase, there is considerable geographic variation [15]. The presence of carbapenemases is associated with increased mortality and treatment failure [11,12,16]. Similarly, delayed appropriate antimicrobial treatment has also been associated with worse outcomes [13,14]. A rapid identification of CR-GNB and their antimicrobial susceptibility pattern is essential to improve outcomes [14]. Several diagnostic methods for CR-GNB have been developed and include modified and EDTA-modified carbapenem inactivation methods (mCIM and eCIM, respectively), hydrolytic methods (e.g., Carba NP-test^®^), and matrix-assisted laser desorption-ionization time of flight mass spectrometry (MALDI-TOF) technology. The turnaround time for each detection test is variable and may be prolonged. Because colony growth in agar plates is warranted, carbapenemase detection may take up to 72 to 96 h when using inactivation methods and MALDI-TOF [17]. To enhance the rapid detection of CR, novel lateral flow immunoassays (e.g., NG-Test^®^ CARBA 5, and O.K.N.V.I. RESIST-5^®^) which detect carbapenemase enzymes using specific antibodies have been recently developed. Lateral flow immunoassays may yield results in 15 min. Among lateral flow immunoassays, NG-Test^®^ CARBA 5 has been reported to more accurately detect MBL [18]. The NG-Test^®^ CARBA 5 (NG Biotech, Guipry-Messac, France) is an immunochromatographic test capable of detecting the five most reported carbapenemases (KPC, OXA-48-like, VIM, IMP, and NDM), with results being available in 15 min [19]. The test is performed using colonies grown in a variety of solid culture media and has been validated for Enterobacterales and *P. aeruginosa* [12,16]. When comparing with genotypic profiling, NG-Test^®^ CARBA 5 using isolated colonies recovered on solid media has 100% and 99% sensitivity and specificity, respectively [20].

Direct NG-Test^®^ CARBA 5 in positive blood culture samples has been described but has not been widely validated [19,20,21,22,23]. Direct testing of positive blood cultures could reduce the time to detection of organisms harbouring carbapenemases and could have favourable impacts in patient outcomes. In this study, we aimed to validate and report on the diagnostic yield of a novel method for carbapenemase detection in positive blood culture vials using NG-Test^®^ CARBA 5.

## 2. Results

A total of 32 isolates were tested. Twenty-six artificially inoculated blood culture bottles and six positive blood culture samples from hospitalized patients were used. Twenty-two Enterobacterales (11 *Escherichia coli*, 5 *Klebsiella pneumoniae*, 3 *Enterobacter cloacae*, 1 *Klebsiella aerogenes*, and 1 *Raoultella ornithinolytica*) and 11 isolates of *P. aeruginosa* were analysed. PCR testing detected 23 carbapenemases in 17 of the isolates (10 NDM, 7 OXA-48, 3 KPC, 2 GES, and 1 VIM) (Table 1).

The mCIM detected 15 carbapenemase harbouring isolates, and among those, the eCIM identified MBL in six. The NG-Test^®^ CARBA 5 in blood detected 20 carbapenemases in 15 isolates (9 NDM, 7 OXA-48-Like, 3 KPC, 1 VIM). In five of these isolates, two enzymes were simultaneously detected.

When comparing the mCIM and eCIM tests with PCR testing, agreement was demonstrated in 24 of 32 samples (75%). The sensitivity, specificity, positive predictive value, and negative predictive value of the mCIM and eCIM test were 53%, 100%, 100%, and 65%, respectively. When comparing the results of the NG-Test^®^ CARBA 5 in blood with mCIM and eCIM, agreement was observed in 26 of 32 samples (81%). The sensitivity, specificity, positive predictive value, and negative predictive value of the NG-Test^®^ CARBA 5 in blood were 100%, 74%, 60%, and 100%, respectively. When comparing the results of the NG-Test^®^ CARBA 5 in blood with PCR testing, agreement was observed in 31 of 32 (97%) samples. In the only recorded disagreement, the carbapenemase that was not detected using the NG-Test^®^ CARBA 5 in blood was an NDM MBL. The sensitivity, specificity, positive predictive value, and negative predictive value of the NG-Test^®^ CARBA 5 in blood were 93%, 100%, 100%, and 94%, respectively (Table 2). Complete antimicrobial susceptibility results and agreement are summarized in Appendix A.

## 3. Discussion

In this study, we validated and report the diagnostic yield of an investigator-developed novel method that can directly detect carbapenemases in positive blood cultures. Our method using NG-Test^®^ CARBA 5 in blood had an excellent diagnostic yield when compared to genotypic profiling. Additionally, our method had a better performance than the widely recommended phenotypic methods (mMIC and eMIC). The latter can be explained by our method’s capability of detecting two or more carbapenemases simultaneously. Isolates harbouring multiple carbapenemases have been reported [24,25], and the detection of simultaneous enzymes could play a decisive role in the choice of treatment. The simultaneous presence of more than one carbapenemase has been reported in 3 to 86% of CR isolates [26,27], depending on the geographic region and the detection methods used.

Previous studies have evaluated the NG-Test^®^ CARBA 5 diagnostic performance when applied to blood samples, and variable sensitivity (97–100%) and specificity (66–100%) have been reported [3,20,22,24]. Stokes et al. [22] reported on a similar method, and our results are in accordance with those reported; nevertheless, the feasibility of the method reported by Stokes et al. could be limited by the fact that their protocol includes a MALDI-TOF-compatible bacterial pellet preparation. Kriger et al. also reported a method for the detection of CR-GNB using the NG-Test^®^ CARBA 5 in blood culture samples, but the method description is brief, limiting its reproducibility [24]. Also, our method requires less material than that proposed by other researchers. In comparison with the method reported by Takissian et al. [20], we did not use Triton X-100. When trying to implement the latter method, we observed that the bubbles generated when adding Triton X-100 caused insufficient absorption when added to the sample well of the test. Although Giordano et al. suggested a method that reported similar results to ours [3], we consider that our use of simple bacterial pellets obtained from positive blood cultures could increase the reproducibility and allow further testing if necessary.

Our study presents limitations that must be acknowledged. NG-Test^®^ CARBA 5 does not detect GES carbapenemases, which is common in *P. aeruginosa* [28]. Similarly, its use is not recommended when studying *Acinetobacter* or *Achromobacter* species. The latter is further reinforced by the fact that carbapenem resistance is not commonly due to the enzymatic mechanisms detected by the NG-Test^®^ CARBA 5. Although our centre is a referral hospital, our results may be representative of the regional epidemiology of carbapenemases in our region [7]. Interpretation of the results must consider that none of the isolates harboured IMP carbapenemases, which could be explained due to their low frequency [3,24]. False negative results for the detection of IMP-14 using the NG-Test^®^ CARBA 5 have been reported [22]. We recognize the need to assess the turnaround time and clinical impact of our method. Our study has several strengths such as the use of recommended genotypic and phenotypic methods as the standard of detection of carbapenemases, and the use of isolates from clinical samples. Additionally, our method detects the simultaneous presence of carbapenemases.

Although the turnaround time for results and the impact of our method were not systematically studied, we expect that our method could play a beneficial role in the care of patients with infections due to carbapenem-resistant Gram-negative bacilli. Because infections due to carbapenem-resistant organisms are increasing, further investigations must focus on establishing which patient’s blood samples should be prioritized for implementing our method. Previously colonized or previously carbapenem-exposed patients with bacteriemia could be the ideal candidates for the test. Rapid detection of carbapenemases results in favourable outcomes and should be implemented. Our results support the use of NG-Test^®^ CARBA 5 for the rapid detection of carbapenemases.

## 4. Materials and Methods

We conducted a single-centre diagnostic test study including CR Enterobacterales and *P. aeruginosa* isolates. Genus and species identification was done using MALDI-TOF-MS (Bruker Daltonics^®^, Bremen, Germany), and antimicrobial susceptibility was obtained using VITEK-2^®^ (BioMérieux, Marcy L’Etoile, France). mCIM, eCIM, and minimum inhibitory concentration (MIC) determination by broth microdilution were performed for all CR isolates according to Clinical & Laboratory Standards Institute (CLSI) [29]. CLSI-recommended MIC breakpoints were considered. We considered CR when non-susceptibility to at least one carbapenem was documented. As a reference standard for carbapenemase identification, polymerase chain reaction (PCR) (Integrated DNA Technologies, Coralville, IA, USA) [30] was used.

Consecutive CR isolates were selected for testing. We used spiked and clinical samples. Regarding the spiked samples, previously characterized CR Enterobacterales and *P. aeruginosa* isolates from different consecutive clinical samples were plated in blood agar. A bacterial suspension with a 0.5 (0.57 ± 0.02) McFarland turbidity scale using sterile water was constituted, of which 1 mL was inoculated in sterile aerobic blood culture bottles (BD^®^, Franklin Lakes, NJ, USA) previously inoculated with 10 mL of sterile blood from the investigators (D.M.-R., B.A.M.-G., S.R.-L., K.M.T.-T., C.M.R.-M., and M.F.G.-L.). Bottles were incubated until positivity using a BD BACTEC FX Instrument^®^ (BD, Franklin Lakes, NJ, USA). Regarding the clinical samples, positive blood cultures from patients with bacteraemia due to CR Enterobacterales and *P. aeruginosa* were used. No polymicrobial samples were used.

The following steps for direct testing of positive blood culture vials using NG-Test^®^ CARBA 5 were followed: (1) 3 mL from the positive blood culture vial were extracted into a 5 mL clot activator/polymer gel-added BD Vacutainer SST^®^ tube and further centrifugated at 10,000 rpm for 10 min, (2) the supernatant was discarded, and the bacterial pellet resuspended in 100 µL (4 drops) of the NG-Test^®^ CARBA 5 buffer, (3) the mixture was homogenized using a vortex mixer, (4) using the NG-Test^®^ CARBA 5 kit-provided pipette, 100 µL of the prepared mixture were added to the test’s sample well (Table 3).

The results were read and recorded 15 min after the process was completed. The described method was investigator-developed. Considering PCR as the gold standard, we compared results between PCR, mCIM, and eCIM, and between PCR and direct NG-Test^®^ CARBA 5 in blood. Additionally, a comparison between direct NG-Test^®^ CARBA 5 in blood and mCIM/eCIM using the latter as reference was done. Agreement, sensitivity, specificity, and positive and negative predictive values were calculated. No personal data were used. All experiments and data managing were performed in accordance with the Declaration of Helsinki. The study was approved by the Institutional Review Board (Comité de Investigación & Comité de Ética en Investigación of the Instituto Nacional de Ciencias Médicas y Nutrición Salvador Zubirán, reference number INF-4025-22-23-1). Because of the study’s nature, informed consent was not required by the Institutional Review Board.

## 5. Conclusions

In conclusion, our method using NG-Test^®^ CARBA 5 directly from blood culture samples presented an excellent diagnostic yield when comparing to genotypic profiling and permits an accurate detection of carbapenemases.

## Figures and Tables

**Table 1 antibiotics-13-01105-t001:** Isolates and carbapenemases studied.

Isolate	CPO/n	KPC/n_CPO_	NDM/n_CPO_	OXA-48/n_CPO_	GES/n_CPO_	VIM/n_CPO_	IMP/n_CPO_
*E. coli*, n = 11	9/11	0/9	7/9	5/9	0/9	0/9	0/9
*K. pneumoniae*, n = 5	4/5	3/4	3/4	1/4	0/4	0/4	0/4
*E. cloacae*, n = 3	1/3	0/1	0/1	0/1	1/1	0/1	0/1
*K. aerogenes*, n = 1	0/1	0	0	0	0	0	0
*R. ornithinolytica*, n = 1	1/1	0/1	0/1	1/1	0/1	0/1	0/1
*P. aeruginosa*, n = 11	2/11	0/2	0/2	0/2	1/2	1/2	0/2

CPO: carbapenemase-producing organisms.

**Table 2 antibiotics-13-01105-t002:** Diagnostic yield of direct NG-Test CARBA 5^®^ and mCIM/eCIM when comparing to polymerase chain reaction.

Method	Sensitivity	Specificity	Agreement
Direct NG-Test CARBA 5^®^ method	93%	100%	97%
mCIM/eCIM method	53%	100%	75%

mCIM/eCIM: modified carbapenem inactivation/EDTA-modified carbapenem inactivation.

**Table 3 antibiotics-13-01105-t003:** Steps for direct testing using NG-Test^®^ CARBA 5.

	Steps
1	When positive, perform a direct Gram stain to ensure only one bacterial morphology is present
2	Extract 3 mL from the positive blood culture vial
3	Add the extracted blood to a 5 mL clot activator/polymer gel-added BD Vacutainer SST^®^ tube
4	Centrifugate at 10,000 rpm for 10 min
5	Discard the supernatant
6	Resuspend the bacterial pellet resuspended in 100 uL (4 drops) of the NG-Test^®^ CARBA 5 buffer
7	Homogenize using a vortex mixer
8	Add 100 µL of the prepared mixture to the test’s sample well
9	Read and interpret the results after 15 min

## Data Availability

Dataset available on request from the authors.

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
