# Peer review of "Rapid Detection of Carbapenemases Using NG-Test® CARBA 5 in Positive Blood Cultures: A Diagnostic Test Study"

_antibiotics, 2024, doi:10.3390/antibiotics13111105_

Round 1
Reviewer 1 Report
Comments and Suggestions for Authors
1. Detail MICs of different carbapenems should be presented.
2. Detail comparison results of mCIM/eCIM with CARBA 5 should be presented, not only summary table of sensitivity, specificity and agreement.
Author Response
On behalf of the authors, I would like to thank you for your valuable review and comments, which undoubtedly improved the quality of our article.
- Detail MICs of different carbapenems should be presented.
Answer: We fully agree with the importance of reporting susceptibility data. Given that the article format is a brief report we have uploaded a supplementary material document that includes the susceptibility data and the agreement between tests. The supplementary material is now referenced in the results section (line 107-109).
- Detail comparison results of mCIM/eCIM with CARBA 5 should be presented, not only summary table of sensitivity, specificity, and agreement.
Answer: We fully agree with the importance of reporting all comparisons. In agreement with the previous answer, the table S1 in supplementary material includes the detailed comparison results.
Reviewer 2 Report
Comments and Suggestions for Authors
Are the results clearly presented ?
The susceptibility profile of the isolates was not presented.
Author Response
On behalf of the authors, I would like to thank you for your valuable review and comments, which undoubtedly improved the quality of our article.
- The susceptibility profile of the isolates was not presented.
Answer: We completely agree with the importance of reporting complete susceptibility data and the comparison results. Given that the article format is a brief report we have uploaded a supplementary material document that includes the susceptibility data and the agreement between tests. The supplementary material is now referenced in the results section (line 107-109).
Reviewer 3 Report
Comments and Suggestions for Authors
Dear authors,
Thank you for your effort,
I would like to highlight some issues that should be addressed:
1. Writing: The manuscript should be revised properly and follow the style of citation according to mdpi.
a. Proofreading is required, there are many examples. Names of microorganisms should be italic (for instance, line 104, 176)
b. Rephrase certain sentences. Example Lines 163, 173
Table 2, footnote: cabapenemase
c. Line 162: hyphen is missed.
d. References, reference 22,
e. The novelty of this work should be explained in introduction and discussion. Add adequate discussion to validate your point
f. Elaborate in discussion point: For example, Line 164: the authors reported previous work by mentioning the names of the author without focusing on the point of comparison (for instance, what are the reagent and steps and the results). Line 169.
g. How did the authors exclude the propriety of the instruments regarding Acinetobacter and Achromobacter (add your justification beside the instrument protocol? Line 177
Best
Author Response
On behalf of the authors, I would like to thank you for your thorough and valuable review. Your comments and suggestions have undoubtedly improved the quality of our article. Additionally. Please note that, as the section order has been modified according to editing requirements.
- Writing: The manuscript should be revised properly and follow the style of citation according to mdpi.
Answer: Thank you for your comment. We have revised the manuscript to ensure citation style.
- Proofreading is required, there are many examples. Names of microorganisms should be italic (for instance, line 104, 176).
Answer: Thank you for your thorough review. We have proofread the manuscript and implemented style corrections. These changes have been implemented in lines 137 and 179.
- Rephrase certain sentences. Example Lines 163, 173
Answer: Thank you for your comments; these sentences needed to be rephrased. We have done so in lines 123-126 and 132-135.
- Table 2, footnote: carbapenemase
Answer: Thank you. We have corrected the footnote in table 1 (please note that table numbering has also changed according to the changes in section order).
- Line 162: hyphen is missed.
Answer: we have corrected this typo. Thank you.
- References, reference 22.
Answer: We have checked reference style and have further discussed this reference in lines 123-126. Thank you.
- The novelty of this work should be explained in introduction and discussion. Add adequate discussion to validate your point.
Answer: Thank you for your encouragement. We have highlighted the importance of this work in the introduction and discussion sections in lines 79-81 and 151-152, respectively.
- Elaborate in discussion point: For example, Line 164: the authors reported previous work by mentioning the names of the author without focusing on the point of comparison (for instance, what are the reagent and steps and the results). Line 169.
Answer: We have rephrased the referenced sentences (see previous answrs) to further clarify our discussion to the readers. Additionally, we have highlighted in lines 130-132, in which we discuss the problems that we observed when testing a different reagent.
- How did the authors exclude the propriety of the instruments regarding Acinetobacter and Achromobacter (add your justification beside the instrument protocol? Line 177.
Answer: We agree with your suggestion. We have further justified the exclusion of Acinetobacter and Achromobacter in lines 138-140.